# Mineralization of 3D Osteogenic Model Based on Gelatin-Dextran Hybrid Hydrogel Scaffold Bioengineered with Mesenchymal Stromal Cells: A Multiparametric Evaluation

**DOI:** 10.3390/ma14143852

**Published:** 2021-07-09

**Authors:** Federica Re, Luciana Sartore, Elisa Borsani, Matteo Ferroni, Camilla Baratto, Allia Mahajneh, Andrew Smith, Kamol Dey, Camillo Almici, Pierangelo Guizzi, Simona Bernardi, Guido Faglia, Fulvio Magni, Domenico Russo

**Affiliations:** 1Bone Marrow Transplant Unit, Department of Clinical and Experimental Sciences, University of Brescia, ASST Spedali Civili, Piazzale Spedali Civili 1, 25123 Brescia, Italy; federicare91@gmail.com (F.R.); simona.bernardi@unibs.it (S.B.); 2Centro di Ricerca Emato-Oncologica AIL (CREA), ASST Spedali Civili, Piazzale Spedali Civili 1, 25123 Brescia, Italy; 3Department of Mechanical and Industrial Engineering, University of Brescia, Via Branze 38, 25123 Brescia, Italy; luciana.sartore@unibs.it (L.S.); k.dey@unibs.it (K.D.); 4Division of Anatomy and Physiopathology, Department of Clinical and Experimental Sciences, University of Brescia, Viale Europa 11, 25123 Brescia, Italy; elisa.borsani@unibs.it; 5Department of Civil, Environmental, Architectural Engineering and Mathematics (DICATAM), University of Brescia, Via Valotti 9, 25123 Brescia, Italy; matteo.ferroni@unibs.it; 6CNR-IMM Bologna, Via Gobetti 101, 40129 Bologna, Italy; 7PRISM Lab, CNR-INO, 25123 Brescia, Italy; camilla.baratto@unibs.it (C.B.); guido.faglia@unibs.it (G.F.); 8Clinical Proteomics and Metabolomics Unit, Department of Medicine and Surgery, University of Milano-Bicocca, Via Raoul Follereau 3, 20854 Vedano al Lambro, Italy; allia.mahajneh@unimib.it (A.M.); andrew.smith@unimib.it (A.S.); fulvio.magni@unimib.it (F.M.); 9Department of Applied Chemistry and Chemical Engineering, Faculty of Science, University of Chittagong, Chittagong 4331, Bangladesh; 10Laboratory for Stem Cell Manipulation and Cryopreservation, Department of Transfusion Medicine, ASST Spedali Civili, Piazzale Spedali Civili 1, 25123 Brescia, Italy; camillo.almici@asst-spedalicivili.it; 11Orthopedics and Traumatology Unit, ASST Spedali Civili, Via Papa Giovanni XXIII 4, 25063 Gardone Val Trompia, 25123 Brescia, Italy; pieroguizzi@tiscali.it; 12Department of Information Engineering (DII), University of Brescia, Via Branze 38, 25123 Brescia, Italy

**Keywords:** bone regeneration hydrogel scaffold, mesenchymal stromal cells, human platelet lysate, Raman spectroscopy, MALDI-MS

## Abstract

Gelatin–dextran hydrogel scaffolds (G-PEG-Dx) were evaluated for their ability to activate the bone marrow human mesenchymal stromal cells (BM-hMSCs) towards mineralization. G-PEG-Dx1 and G-PEG-Dx2, with identical composition but different architecture, were seeded with BM-hMSCs in presence of fetal bovine serum or human platelet lysate (hPL) with or without osteogenic medium. G-PEG-Dx1, characterized by a lower degree of crosslinking and larger pores, was able to induce a better cell colonization than G-PEG-Dx2. At day 28, G-PEG-Dx2, with hPL and osteogenic factors, was more efficient than G-PEG-Dx1 in inducing mineralization. Scanning electron microscopy (SEM) and Raman spectroscopy showed that extracellular matrix produced by BM-hMSCs and calcium-positive mineralization were present along the backbone of the G-PEG-Dx2, even though it was colonized to a lesser degree by hMSCs than G-PEG-Dx1. These findings were confirmed by matrix-assisted laser desorption/ionization mass spectrometry imaging (MALDI-MSI), detecting distinct lipidomic signatures that were associated with the different degree of scaffold mineralization. Our data show that the architecture and morphology of G-PEG-Dx2 is determinant and better than that of G-PEG-Dx1 in promoting a faster mineralization, suggesting a more favorable and active role for improving bone repair.

## 1. Introduction

Regenerative medicine aims to induce self-repair (regeneration) of tissues and organs by unconventional and advanced approaches such as bioengineering of three-dimensional (3D) scaffolds with human mesenchymal stromal cells (hMSCs) [1]. Usually, bone substitute materials are required in critical-sized bone defect repair; nevertheless, they do not meet the clinical requirements of biodegradability, structural support and osteoinductive property [2]. Various strategies have been explored to overcome these critical aspects via bone tissue engineering approaches that incorporate biomimetic scaffolds as a novel platform for phenotypically stable tissue formation and stem cell differentiation. 

Scaffolds designed as 3D porous biodegradable substrates promote easier cell-biomaterial interactions as well as the transport of gases, nutrients and regulatory factors for cell survival, proliferation and differentiation. 

In particular, for bone regeneration, they should also mimic the 3D bone structure in terms of physical mechanical properties as well as osteoinductive (bone inducing), osteoconductive (bone supporting) and osteogenic (bone forming) features [3]. The majority of scaffolds that are currently used for bone regeneration applications are natural polymers (e.g., chitosan, hyaluronic acid and collagen), synthetics polymers (e.g., polylactic acid (PLA), polycaprolactone (PCL)) or bioactive ceramics (e.g., hydroxyapatite). The hybrid hydrogels, combining the benefits of natural and synthetic polymers, have attracted consideration as support of hMSCs which are able to differentiate into cells of the mesodermal lineages and other embryonic lineages, including osteocytes and chondrocytes, among others [4,5,6]. 

A critical parameter to evaluate the tissue-engineered constructs is the identification of a quality control system to allow the qualitative and quantitative measurement of the mineralization area, which is characterized by the deposition of a modified form of hydroxyapatite (carbonated apatite) that is comparable to the mineral composition of natural bone [7]. Hydroxyapatite is composed of calcium and phosphate, with a Ca/P ratio close to 1.67. For this reason, Ca^2+^ and PO4^3−^ are the most important ions for mineralization [8,9].

Commonly, Von Kossa or Alizarin Red histochemistry characterization is used as a validated step to ensure calcium deposits [10,11,12]. Scanning Electron Microscopy (SEM) evaluation, together with Energy Dispersive X-ray (EDX) spectroscopy, is also chosen as a means of visualizing the scaffold surface and to quantify individual elements of calcium and phosphorus commonly present in mineral deposits (e.g., [13]). In addition, one of the increasingly applied technologies is represented by micro-computed tomography (micro-CT) to characterize and visualize scaffolds in 3D [11]. Despite these efforts, the mineralization of scaffolds has not been deeply investigated using this technology, which is more frequently applied in the chemical and engineering fields. 

Another technique typical of material science that was applied in literature for the study of mineralization and osteogenic differentiation of hMSCs is Raman spectroscopy (RS). This is a non-destructive technique which measures the vibrational spectrum, allowing discrimination between the chemical species. This means that RS allows measuring protein contribution of the cultured cells and the phosphate bands’ contribution to mineralization due to the presence of hydroxyapatite [14], with a spatial resolution of the order of a few microns and poor interference from water signal [15].

The microscopic investigation of non-flat surfaces by means of optical microscopy of the Raman system suffers from low depth of focus, typical of 50× objective, but is efficiently guided by the visualization of the scaffold surface and localization of mineral deposits obtained by SEM, and also from the conventional staining method (Von Kossa). 

Another innovative possibility is the use of matrix-assisted laser desorption/ionization mass spectrometry imaging (MALDI-MSI) is an advanced method used to map the distribution of biomolecules without any probes. However, hydroxyapatite crystals of the bone tissue make it problematic to determine the distribution of biomolecules using MALDI-MSI [16] and this explains why the few studies reported in the literature focused on the lipidome and metabolome of MSCs. However, its use for detecting mineralization and molecular indicators of differentiation in regenerative scaffolds is not widespread, representing an area that has been relatively untapped by this technique and may provide some novel molecular insights into the osteogenic process. Moreover, recent advancements in this field have enabled in situ lipodomic imaging by MALDI-MS to provide near single-cell lateral resolution and can be a highly appropriate tool to map the molecular heterogeneity in different clusters of hMSCs present within our 3D regenerative scaffold [17]. 

In this work, we investigated the capability of novel three-dimensional gelatin–dextran hybrid hydrogel scaffolds (G-PEG-Dx) having different architecture and morphology (i.e., degree of crosslinking and pore size) to support mineralization in BM-hMSCs cultured with growing medium (GM) or osteogenic medium (OM) combined with hPL. Consolidated methodologies, such as Von Kossa histochemistry via optical microscopy and SEM-EDX, as well as innovative technologies such as RAMAN spectroscopy and MALDI-MSI, were used in order to analyze the distribution, morphological characteristic and composition of mineralized area in 3D scaffolds. 

## 2. Materials and Methods

### 2.1. Reagents

Type A gelatin (G) (pharmaceutical grade, 280 bloom, viscosity 4.30 mPs, produced from pig skin) was purchased from Italgelatine, Cuneo, Italy. Poly(ethylene glycol) diglycidyl ether (PEGDGE) (molecular weight 526 Da) and Dextran (D) (molecular weight 70,000 Da) were purchased from Sigma-Aldrich Co, Milan, Italy. Aminated dextran (D-NH2) (molecular weight = 70,000 Da and degree of polysaccharide ring amination = 30–40%) was prepared starting from the same dextran following a procedure described below. In addition, 4-Nitrophenyl chloroformate and ethylene diamine were purchased from Fluka, Milano, Italy.

DMEM, L-glutamine, penicillin-streptomycin, sodium pyruvate, 4′,6-diamidino-2-phenylindole (DAPI), xylene, ethanol, Paraplast and poly-l-lysine were purchased from Sigma-Aldrich, St. Louis, MO, USA. Amphotericin B and MEM Non Essential Amino Acids Solution were purchased from Gibco, Thermo Fisher Scientific, Madison, WI, USA. Hematoxylin–eosin stain (H&E) and Von Kossa stain were purchased from Bio-Optica, Milan, Italy. Masson–Goldner Trichrome stain was purchased from Merck KGaA, Darmstadt, Germany. 

### 2.2. Synthesis of G-PEG-Dx1 and G-PEG-Dx2 Hydrogels

G/PEG/Dx1 hydrogel was prepared in aqueous solution, and the synthetic procedure involved the reaction mainly between gelatin amino-groups and the epoxy groups of poly(ethylene glycol)diglycidyl ether (PEGDGE) without using any additives or catalysts. Briefly, gelatin (6 g) was dissolved in 60 mL distilled water at 45 °C under mild magnetic stirring followed by dropwise addition of PEGDGE (1.4 g). The reaction mixture was continually stirred, gently, for some minutes and a previously prepared D solution (7 wt.%, 25 g) was then added into it. Afterwards, the reaction mixture was gently stirred at 45 °C for 45 min and poured into the glass plate at room temperature for gel formation. The resulting gel was carefully peeled off and cut into rectangular bars (5 cm × 1 cm × 1 cm) and placed into a Pyrex crystallizing dish. The freezing was carried out by resting the crystallizing dish on the surface of a 3-centimeter-deep pool of liquid nitrogen, enabling freeze-casting at −196 °C. Freezing was assessed visually. The gel was incubated for 30 min at the freezing temperature to ensuring complete freezing. Subsequently, the frozen gel was transferred to the lyophilizer, operating under vacuum at −60 °C, for sublimation of ice crystals, resulting in a porous gel. The dry porous gel was post-cured at 45 °C for 2 h in the oven under vacuum to complete crosslinking reaction. The gel was washed several times with distilled water at 37 °C to eventually remove the unreacted reagents as well as soluble components and finally freeze-dried in lyophilizer.

Similarly, G-PEG-Dx2 hydrogel was prepared following the identical condition, with the exception of adding D-NH_2_ solution (7 wt.%, 25 g) instead of D solution.

To obtain D-NH_2_, the hydroxyl groups of polysaccharide ring were reacted with 4-nitrophenyl chloroformate, a common and versatile coupling reagent for the activation of alcohols. The resulting phenyl carbamate derivatives was coupled with ethylene diamine to obtain D-NH_2_. The degree of amination, evaluated by titration, was between 30 and 40% of the total polysaccharide rings.

One hundred cubic samples (5 × 5 × 4 mm^3^) of each hydrogel compositions were cut by mechanical saw in dry state and packed into polypropylene bag and sealed under vacuum. Packed hydrogel scaffold samples were sterilized by gamma irradiation with Cobalt 60 g rays using 27–33 kGy following UNI EN ISO 11137 (Sterilization of Health Care Products) [18].

### 2.3. Morphological and Mechanical Characterization of Hybrid Hydrogels

Compression tests were carried out by an Instron series 3366 testing machine (INSTRON, Norwood, MA, USA), equipped with a 50 N load cell. The specimens were tested at room temperature after immersion in distilled water for 24 h at 37 °C (swollen condition). Samples were cut into cylindrical specimens and their dimension was measured using an optical travelling microscope. Samples were compressed at a strain rate of 10%/min up to 50% strain, then immediately unloaded. Before commencing the compression test, a load of 0.01 N was applied to ensure complete contact between the sample surface and plate. At least six specimens were tested for each hydrogel composition. 

Engineering stress (σ) was calculated by dividing the recorded force by the initial cross-sectional area. Engineering strain (ε) under compression was defined as the change in height relative to the original height of the freestanding specimen. The initial Young Modulus (stiffness) was calculated from the slope of the compressive stress–strain curves within the range of 5–10% strain. The compressive strength was defined as the stress at 50% strain.

The texture, morphology and porous structure of hydrogels were observed in both parallel and perpendicular directions using a stereomicroscope (LEICA DMS 300, Leica Microsystems GmbH, Wetzlar, Germany) under reflected light illumination.

### 2.4. Swelling Ratio and Hydrolytic Mass Loss Evaluation

The dry weighed (Wi) samples were incubated in distilled water at 37 °C over a three-week period. At regular intervals of 1, 10 and 21 days, the samples were removed, weighed (Ww), rinsed with fresh water, air dried followed by vacuum dried at 45 °C for 4 h, and finally dried mass (Wf) was measured. The swelling ratio (%) and mass loss (%) were calculated using following equations:Swelling ratio (%) = (Ww − Wf)/Wf × 100(1)
Mass loss (%) = (Wi − Wf)/Wi × 100(2)

### 2.5. Human Platelet Lysate Production 

Human Platelet Lysate (hPL) for MSCs expansion was obtained from blood donations belonging to the Blood Bank of ASST Spedali Civili of Brescia, Italy and produced according to standardized clinical grade procedures in closed systems [19] and as previously described [13]. For the technical procedure’s details, please refer to F. Re et al. [13].

### 2.6. Human Bone Marrow Mesenchymal Stromal Cells (BM-hMSCs) Culture

Human Bone Marrow Mesenchymal Stem Cells (BM-hMSCs) were purchased from PromoCell, 69126 Heidelberg, Germany, and expanded as previously described [13]. Briefly, BM-hMSCs were expanded in the presence of a growth medium (GM), a high glucose-based Dulbecco’s modified Eagle’s medium (DMEM) with 2% L-glutamine/penicillin-streptomycin/amphotericin B solution, 1 mM sodium pyruvate, MEM Non Essential Amino Acids Solution 1X and 10% fetal bovine serum (FBS) (referred as complete medium FBS) or 5% human platelet lysate (hPL) (referred as complete medium hPL) at 37 °C and 5% CO_2_ in an incubator. 

### 2.7. BM-hMSCs Cultured in Osteogenic Medium

A cell suspension at a cellular density of 10^6^ cells/mL (36 × 10^3^ cells/scaffold) was added to the scaffolds for osteogenic differentiation in static conditions as previously described [13]. For the technical procedure’s details, please refer to F.Re et al. [13]. All samples were analyzed in triplicates. After 28 days of culture, scaffolds were fixed with paraformaldehyde 4% for 1 h at 4 °C and dehydrated for subsequent analysis. 

### 2.8. Histomorphological Analysis

Scaffolds were embedded in paraffin (Paraplast, Sigma, St. Louis, MO, USA) according to standard procedures. Serial sections (8 um thick) were cut by a microtome (Microm HM 325, Thermo Scientific, Walldorf, Germany) and collected on poly-l-lysine-coated glass slides. Sections were deparaffinized in xylene, rehydrated and stained with hematoxylin–eosin stain (H&E) for general morphology (Bio-Optica, Milan, Italy); Masson–Goldner Trichrome stain (Bio-Optica) was performed to identify collagen fibers in light green and Von Kossa stain to evaluate the presence of calcium deposits in black/brown color (Bio-Optica) following the manufacturing staining protocols. Moreover, 4′,6-diamidino-2-phenylindole (DAPI) fluorescent stain for cell nuclei was performed to confirm the presence of cells in scaffolds using ZEISS Observer Z1 fluorescence microscope (Zeiss, Oberkochen, Germany). In order to quantify the calcium deposits, digitally fixed images (arbitrary standardized area) for each section (five serial sections/sample) were analyzed by optical light microscope (Olympus BX50, Olympus, Hamburg, Germany). The analyses were performed by two blinded investigators using a camera equipped with an image analysis system (Image-Pro Premier 9.1; 2018, Immagini e Computer, Milan, Italy). The following parameters on calcium deposits were measured: (1) percentage of positive area within scaffold meshes, (2) intensity (luminosity-lum), in which the highest value represents the less intense staining, and (3) the mean diameter.

### 2.9. Scanning Electron Microscopy Analysis

A longitudinal section of the dry samples was exposed by razor cut and observed at the SEM with no additional preparation. 

The samples were mounted with conductive tape and the ZEISS EVO LS 10 (Carl Zeiss AG, Oberkochen, Germany) environmental SEM was operated at 20 keV and in the 0.1–0.01 mbar pressure range to minimize the electrostatic charging. Morphological and compositional images were obtained using the Backscattered Electrons detector, while the local changes in elemental composition were investigated using the Energy Dispersive X-ray (EDX) analyzer. 

### 2.10. Raman Spectroscopy

Spectroscopic analysis was performed with a micro-Raman modular system by Horiba (Horiba, Kyoto, Japan), equipped with single monochromator (iHR320) and Peltier cooled CCD camera (Horiba, Kyoto, Japan). The acquisition was performed with a 50× long working distance objective with Numerical Aperture (NA) 0.55, to allow investigation of samples on scaffolds cut in pieces. A spectral resolution of 1.68 cm^−1^ was provided by 600 g/mm grating. IR excitation at 785 nm by a solid state laser allowed us to minimize the fluorescence pollution of Raman spectra; nonetheless, fluorescence background is still observed in G-PEG-Dx scaffolds and in cultured scaffolds with cells. The system was calibrated using a silicon substrate at 520.5 cm^−1^. The spatial resolution is usually higher than the theoretical diffraction limited spatial resolution, obtained by the Airy formula (1.22 λ/NA = 1.7 μm), due to laser scattering or photons interactions inside the sample. The investigated spectral range is between 150 and 1800 cm^−1^. For each spectral scan, laser exposure time was set to 120 s with 4 accumulations. 

### 2.11. Matrix-Assisted Laser Desorption/Ionization Mass Spectrometry Imaging (MALDI-MSI)

#### 2.11.1. Sample Preparation

Ten-micron-thick sections were cut for G-PEG-Dx1 (OM+HPL) and G-PEG-Dx2 (OM+HPL), then mounted onto conductive glasses coated with indium tin oxide (Bruker Daltonik GmbH, Bremen, Germany). The slides were then placed in a desiccator for 30 min and analyzed immediately. 

Then, 10 mg/mL 9-aminoacrdine (9-AA) were dissolved in a 70% methanol solution and deposited using the HTX TM-Sprayer™ (HTX Technologies, LLC, Chapel Hill, NC, USA), with the following parameters: temperature 85 °C; number of passes 6; flow rate 0.2 mL/min; velocity 1100 mm/min; track spacing 2 mm; pressure 10 psi [20].

#### 2.11.2. MALDI-MS Parameters

All analyses were performed using a rapifleXTM MALDI TissuetyperTM mass spectrometer (Bruker Daltonics, Bremen, Germany) equipped with a Smartbeam™ 3D laser (Bruker Daltonik GmbH, Bremen, Germany). External calibration was performed using red phosphorus clusters in the m/z range of 0 to 2000. Mass measurements in negative ion reflectron mode were acquired in the m/z range of 500 to 1000. Two hundred shots were accumulated for each spectrum and the matrix suppression deflection was set to *m*/*z* 400. The samples were rastered at a lateral resolution of 20 × 20 (x,y) µm with a laser scan range of 16 µm per pixel. 

For in situ MALDI-MS/MS, a single precursor ion was selected by using the smallest precursor ion selector (PCIS) window possible and dissociated using LID-LIFT™ technology, with the laser energy being set within a range of 40–70%. This process was performed until an MS/MS spectrum was obtained from the accumulation of ~100,000 laser pulses.

#### 2.11.3. Data Processing

Data files containing the individual spectra of each entire measurement region were imported into SCiLS Lab MVS 2021a Pro software (http://scils.de/; Version 2021a. Accessed on February 2021, Bremen, Germany) for spectra pre-processing and to annotate regions of interest (ROIs). The ion distribution for *m*/*z* 885.59 (PI(18:00/20:4)) was used to indicate regions containing cells and guide the annotation. Average spectra were then generated for those cells present in G-PEG-Dx1 (OM+HPL) and G-PEG-Dx2 (OM+HPL). Moreover, Receiver Operative Characteristic (ROC) analysis and the Wilcoxon Rank Sum Test were performed, with an Area Under Curve

(AUC) value ≥0.80 and *p* < 0.05 being required for a lipid ion to be considered as discriminatory.

For lipid annotation of the discriminatory ions, the previously generated average spectra were imported into mMass (version 5.5.0, 2013, open source), where peak picking (S/N ≥ 5) was performed. The peak list was then cross-referenced the LIPID MAPS database, setting a tolerance value of 15 ppm for the peak annotation.

For additional lipid identification, product ions in the acquired MALDI-MS/MS spectra were annotated within mMass after being cross-referenced with known product ions generated by the fragmentation of different phospholipid species. Then, identifications were assigned by matching the mass of the precursor ion with lipids listed in the METLIN, HMDB and Lipid Maps databases.

### 2.12. Statistical Analysis

The data collected using Von Kossa staining were analyzed by one-way ANOVA followed by the Bonferroni test. The levels of significance were set at 5, 1 and 0.1% (*p* < 0.05, 0.01, 0.001).

## 3. Results

### 3.1. Preparation and Characterization of Scaffold G-PEG-Dx1 and G-PEG-Dx2

The design and development of novel scaffolding materials remain a central point of interest in the field of scaffold-based tissue engineering. At first, we synthesized anisotropic dextran-based hybrid hydrogels using the uniaxial freezing technique developed in our laboratory [21]. Hydrogels were prepared in an aqueous solution where grafting and crosslinking reactions mainly occurred between the end epoxide groups of functionalized PEG and free α-amino groups of gelatin chain without using any additives or catalysts. Poly(ethylene glycol)diglycidyl ether (PEGDGE) was selected as the crosslinking agent for the possibility of epoxide groups to react not only with primary amino-groups of gelatin, but also with other available functional groups (e.g., secondary amine, carboxylic or hydroxyl groups). An excess of epoxy-PEG over protein amino-groups was employed aiming to obtain adducts with a number of reactive end groups able to produce crosslinking/grafting. For imparting anisotropy, directional freezing followed by drying approach was adopted for increasing the crosslinking density. In addition, we forced the curing process by heating the dried samples at 45 °C. Two structurally stable G-PEG-Dx hydrogels characterized by identical proportions between the components G, PEG and D and designated hydrogel containing unmodified D as G-PEG-Dx1 and hydrogel containing modified D (D-NH_2_) as G-PEG-Dx2 have been prepared (Scheme 1). G, PEG and D content in the dry samples were 66, 16 and 18 wt.% respectively. Addition of D-NH_2_ into the reaction mixture might increase the crosslinking reaction due to the availability of primary amino-groups in functionalized dextran polymer chains, which could produce hybrid network systems. D-NH_2_ could provide the possibility of obtaining an interpenetrating polymer network hydrogel that is an entangled combination of two cross-linked polymers, Gelatin-PEG and Dextran-PEG, and they cannot be separated unless chemical bonds are broken. This possibility is also favorable as it might improve their biomedical applications and physical-mechanical properties. Furthermore, due to the high degree of amination of D-NH_2_ in which 30–40% of the repeating units of dextran were aminated, a decrease in the length of the lattices (i.e., the distance between the crosslinks) of G-PEG-Dx2 compared to G-PEG-Dx1 was expected. The developed hybrid hydrogels were structurally stable and allowed effective gamma sterilization without compromising their physical-mechanical properties [22]. As shown in Figure 1A, both the dry hydrogels showed a highly interconnected irregular pore morphology and micro-macro spherical pore and channels are homogeneously distributed into the network. Porosity and pore size are believed to promote and regulate tissue regeneration and vascularization, mass transport, including nutrients supply and wastes disposal [23]. Mean pore sizes and ranges of pore dimensions were evaluated on dry hydrogels using a stereomicroscope. Mean pore sizes of the G-PEG-Dx1 and G-PEG-Dx2 hybrid hydrogels were found to be 235 and 115 μm, respectively, as shown in Table 1, and the similar reduced trend was also visible for ranges of pore dimensions. The hydrophilic nature of the component polymers and the porous architecture of the network enabled high water retention ability to the hydrogels, as shown in Figure 1A. 

However, hydrogel G-PEG-Dx1 showed a higher amount of water uptake compared to that of G-PEG-Dx2, and the cause might be due to the lower cross-linking density (as confirmed by mechanical test). Consequently, this higher water uptake and lower crosslinking density resulted in higher hydrolytic mass losses for G-PEG-Dx1 as compared to G-PEG-Dx2 hydrogels, as shown in Figure 1C and Table 1. The mechanical property of the scaffold is a key regulator for driving many cellular behaviors [24]. Figure 1D shows representative compressive stress–strain curves for both hybrid hydrogels under wet conditions. The stress–strain compressive curves obtained for G-PEG-Dx1 and G-PEG-Dx2 hydrogels clearly showed three distinct regions depending on the slope of the curve: (i) linear elastic region due to the bending of pore walls and lamellae (0–20% strain), (ii) collapsed plateau region with plain slope as a result of pore walls and lamellae buckling and yielding (20–40% strain) and (iii) densification region with higher slope (strain-stiffening) due to the pore walls/lamellae crushing together (>40% strain). It was evident that the incorporation of D-NH_2_ into the network increased the compressive Young Modulus (rigidity) of the resulting hydrogel. The stiffness of the G-PEG-Dx1 and G-PEG-Dx2 hybrid hydrogels was found to be 0.26 and 0.42 MPa, respectively, as shown in Table 1. It is reasoned that incorporation of D-NH_2_ into the G-PEG reaction mixture results in greater availability of primary amino-groups (–NH_2_) to react with the epoxy groups of PEG, thus leading to higher crosslinking density (Scheme 1). For the same reason, G-PEG-Dx2 hydrogels showed better structural stability compared to that of G-PEG-Dx1 hydrogels.

### 3.2. Evaluation of Cells Colonization in G-PEG-Dx1, G-PEG-Dx2: Haematoxylin-Eosin Stain (H&E) and Masson-Goldner’s Trichrome Stain

H&E was performed in order to appreciate the scaffold morphology and cell colonization of both hydrogels in different culture conditions at day 28. Since the hydrogels showed affinity for staining, the entire structure of the hydrogels was clearly observable, showing different sized pores. In contrast, it was difficult to appreciate cells (Figure 2) even if was possible to recognize them at higher magnification (Figure 2M,N). Cells were more visible in G-PEG-Dx1 than G-PEG-Dx2 pores because of their larger size. DAPI staining confirmed the presence of cells in both the hydrogels (Figure 3F, Appendix A). Moreover, extracellular matrix (ECM) deposition was observed in G-PEG-Dx1 pores, which was confirmed by Masson–Goldner’s Trichrome stain (Figure 3A–D). Nevertheless, the hydrogels sequestered the colors, and the presence of ECM was not clearly detected in G-PEG-Dx2 (Figure 3E).

### 3.3. Analysis of Scaffold Mineralization in G-PEG-Dx1, G-PEG-Dx2

#### 3.3.1. Von Kossa Stain

Von Kossa stain showed that the mineralization occurred only within scaffold meshes. In particular, the first consideration was that G-PEG-Dx2, with its small pores, presented more possibilities of calcium deposition. Analyzing the percentage positive area of mineralized meshes positive area confirmed this trend. Moreover, the calcium deposits were mainly localized on the scaffolds treated with OM in both G-PEG-Dx1 and G-PEG-Dx2 at day 28 (Figure 4C). Particularly, G-PEG-Dx2 treated with OM+hPL showed more scaffold mineralization compared to G-PEG-Dx1 with the same treatment (G-PEG-Dx1, OM+hPL vs. G-PEG-Dx2, OM+hPL, *p* < 0.01) (Figure 4C). Moreover, there was no statistically significant difference using OM-hPL and OM-FBS in G-PEG-Dx2 (G-PEG-Dx2, OM+hPL vs. G-PEG-Dx2, OM+FBS) (Figure 4E). Initial calcium deposits were observed in both the hydrogels without osteogenic stimuli (GM) and hPL, which appeared more promising compared to FBS, even if a quantification was not possible considering the absence or very low presence of deposits at day 28 (Appendix A). 

Considering the results obtained using osteogenic differentiation media, we then investigated both the calcium deposits’ intensity and mean diameter. An important consideration regarded the distribution of calcium deposits in OM samples. In fact, the mineralization appeared heterogeneous in hydrogels treated with FBS: it was more present and with an intense brown color at the hydrogel periphery, while in the hydrogel core, the deposits appeared in lower numbers and with a lower intensity (Figure 5A,C), with the best performance observed in G-PEG-Dx1 with respect to G-PEG-Dx2. On the other hand, treatment with hPL guaranteed a more homogeneous distribution throughout the entire hydrogel, with the best performance observed in G-PEG-Dx2 with respect to G-PEG-Dx1 (Figure 5B–D). In particular, the results highlighted that a greater mean intensity was present in G-PEG-Dx1 treated with FBS compared to G-PEG-Dx2, OM+hPL (G-PEG-Dx1, OM+FBS vs. G-PEG-Dx2, OM+hPL, *p* < 0.05). G-PEG-Dx2 treated with OM+hPL resulted in a larger intensity of the dark stain with respect to G-PEG-Dx1 treated with OM+hPL (G-PEG-Dx1, OM+hPL vs. G-PEG-Dx2, OM+hPL, *p* < 0.001) and G-PEG-Dx2 treated with OM+FBS (G-PEG-Dx2, OM+hPL versus G-PEG-Dx2, OM+FBS, *p* < 0.001) (Figure 5E).

Moreover, the mean diameter of calcium deposits was unaltered in G-PEG-Dx1 treated with FBS with respect to G-PEG-Dx2 treated with hPL (G-PEG-Dx1, OM+FBS vs. G-PEG-Dx2, OM+hPL, *p* > 0.05) considering the high variability of single diameter of calcium deposits (Figure 5F). In light of the results obtained using optical microscopy, the G-PEG-Dx1 (OM+HPL) and G-PEG-Dx2 (OM+HPL) conditions were used given the differing extents of scaffold mineralization between them.

#### 3.3.2. SEM-EDX Analysis 

Figure 6 shows the morphology of the polymeric scaffold and the evidence of the mineralization. The panoramic view (upper part of the figure) shows the top side of the cylindrical scaffold and the exposed longitudinal section. The Backscattered Electron imaging highlights a decoration of bright particles at the scaffold surface and in the open porosity of the scaffold as well. The view at high magnification (bottom-left of the picture) reveals that the bright particles are indeed aggregates of round particles which measure few microns in size and feature an elemental composition different than the surrounding polymeric scaffold/biological matrix. The EDX measured the elemental composition of these details, which were shown to contain primarily Ca and P, oxygen was also detected but the contribution of the underlying matrix could not be discriminated. The mapping of Ca and P dispersion in the sample is reported in the bottom part of the picture as blue/green colored images. The resulting maps are similar to the magnified SEM image and indicate that both Ca and P, i.e., the cations of hydroxyapatite, are characteristic elements of the bright particles, which can be considered as the result of the penetration of the cellular culture inside the scaffold and evidence of the mineralization process.

#### 3.3.3. Raman Spectroscopy 

We studied the mineralization of hMSCs by Raman spectroscopy, by following the Raman signal of hydroxyapatite in the polymeric scaffold. The polymeric scaffold pieces were first observed with a low magnification objective (10×) to locate regions previously analyzed by SEM (Figure 6). Then, a 50× objective was used to collect the Raman signal. 

Figure 7A shows the collected Raman spectra from the G-PEG-Dx2 OM+hPL sample and the G-PEG-Dx2 scaffold without any cultured cells. The Raman spectra were corrected by subtracting the fluorescence signal. 

Figure 7B,C are optical images with 50× objective from the regions of interested on OM+HPL samples: we selected spherical particles, which were a few microns in diameter, similar to those observed by SEM, in several regions outside and inside the scaffold (B), and a region of the scaffold with no visible mineralization, most frequently found in the inner area (C). The Raman spectrum of the scaffold G-PEG-DX2 shows several vibrations (Figure 7A, black line) assigned in accordance with literature [25,26,27]. The peaks at 854, 1238, 1284 and 1450 cm^−1^ can be assigned to CH_2_ vibration of the PEG. The band at 1445 cm^−1^ is due to CH_2_ bending, while the peak at 1632 cm^−1^ is assigned to the amide I (C=O stretching vibration) of the G-PEG. The Raman spectrum of OM+HPL zone C (Figure 7A, gray line) is characterized by a strong signal with increased fluorescence, promoted by the cultured cells; the band observed at 1450 cm^−1^ and 1632 cm^−1^ could either be assigned to the scaffold signal or protein contribution of the cultured cells. 

The Raman spectrum of OM+hPL zone B, collected on spherical particles (green line), clearly shows the phosphate bands at 960 cm^−1^, characteristic of the ${\mathrm{PO}}_{4}^{3-}$ tetrahedral stretching vibration of hydroxyapatite, in agreement with reports in literature [14]. The small band at 440 cm^−1^ can also be assigned to hydroxyapatite. Some bands at 1450 cm^−1^ and 1632 cm^−1^ are also present and they pertain to the signal of scaffold or cells observed in zone C. Thus, we can affirm that the presence of hydroxyapatite is confirmed by Raman spectroscopy in the area of bright micrometric particles.

#### 3.3.4. MALDI-MSI Analysis

MALDI-MSI was performed in order to investigate the possibility of detecting alterations in the lipid composition of those hMSCs cultured in the G-PEG-Dx1 and G-PEG-Dx2 scaffolds and that may be linked with the degree of mineralization. Initially, the distribution of *m*/*z* 885.59 (PI(18:00/20:4), a lipid signal which has previously being reported to be a potential marker of chondrogenic differentiation [28], was monitored and was shown to be distributed across a wider area in G-PEG-Dx2 with respect to G-PEG-Dx1 (OM+hPL) (Figure 8A) and is supportive of the evidence observed with the previous techniques (Figure 4, Figure 5 and Figure 6). Moreover, when comparing its intensity within the cellular regions from these two scaffolds, it was also shown to be of a higher intensity in G-PEG-Dx2 (OM+hPL), with an AUC value ≥0.80 and *p* < 0.05 (Figure 8B). 

When comparing the hMSCs produced by the two scaffolds with supervised statistical analysis, two lipid signals were shown to be of higher intensity in G-PEG-Dx1 (OM+hPL) while an additional 12 lipid signals were shown to be of higher intensity in G-PEG-Dx2 (OM+hPL), which are indicated in Figure 9A,B. A putative lipid ID could be assigned to five of these lipid signals: PI(18:0/18:1), PG(14:1/20:5), PPA(16:0/18:1), PI(16:1/14:1) and PG(22:6/18:2). These are also indicated in Figure 9.

## 4. Discussion

Scaffolds are three-dimensional porous structures employed in regenerative medicine and tissue engineering for the repair of tissue and organs. Different materials, both synthesized and printed, have been shown to be suitable for cell culture and sustaining the development of native tissue [29,30,31], but only few of them have been deemed able to play a determinant and active role in inducing and modulating the cell proliferation and differentiation.

In our previous studies, we described how the combination of our gelatin–chitosan hybrid hydrogels (G-PEG-Ch) with hMSCs may be a promising strategy for bone and cartilage regenerative medicine using human mesenchymal stromal cells [13,20,21,22]. 

Recently, special attention has been reserved for the development of dextran-based hydrogels that are precisely manipulated with desired structural properties for different therapeutic aims [32]. Here, we designed and developed novel hydrogels based on gelatin, PEG and dextran with identical components concentration but different in architecture and mechanics for bone regeneration, and we tested their biocompatibility, seeding them with BM-hMSCs.

Dextran is a hydrophilic carbohydrate biopolymer that degrades in certain physical environments without any effect on the cell viability and has attracted attention in biological systems thanks to its biocompatibility, biodegradability and ability to favor cell attachment and growth [33].

Different lattice architectures and morphologies have been obtained using aminated dextran as an alternative to dextran. Due to the availability of primary amino groups in dextran polymer chains, D-NH_2_ could produce an interpenetrating polymer network hydrogel which is an entangled combination of two cross-linked polymers, G-PEG and D-PEG. This possibility is also favorable as it could improve their biomedical applications and physical-mechanical properties. Furthermore, due to the high degree of amination of D-NH_2_, a probable decrease in the lattice length (i.e., the distance between the crosslinks) of G-PEG-Dx2 compared to G-PEG-Dx1 was expected. Moreover, the incorporation of D-NH_2_ into the G-PEG network results in an increased availability of reactive primary amino groups to react with the epoxy groups of PEG, increasing the degree of crosslinking and decreasing the size of the pores and channels. The resulting higher crosslinking density leads to higher stiffness and stability as well as lower hydrophilicity of G-PEG-Dx2 with respect to those of G-PEG-Dx1 hydrogels. 

Mineralized biomaterials are promising for use in bone tissue engineering. Physiologically, bone regeneration starts with the differentiation of osteogenic lineage cells from initial mesenchymal progenitor cells to the mature osteocyte in mineralized connective tissue, with the last phase of the tissue process being mineralization of the extracellular matrix (ECM) and establishment of osteocyte concomitant. Generally, almost 4 weeks or more are required for mineralization to occur [34]. Nevertheless, in our previous experience, G-PEG-Ch mineralization was already detectable at 21 days of cells differentiation [13]. Therefore, it is likely that this system is able to trigger the fast production of calcium and phosphorous. For this reason, we studied the seeded G-PEG-Dx with hMSCs, analyzing the morphology, the distribution and composition of the mineralized area and the osteogenic lipidomic profile of the differentiated hMSCs. 

Scaffolds were initially studied from a morphological point of view using optical microscopy techniques. Morphological data showed a different degree of porosity between the two types of hydrogels, with the G-PEG-Dx1 showing larger pores with respect to G-PEG-Dx2. Moreover, the presence of extracellular matrix and cells was clearly present inside the G-PEG-Dx1 pores in all treatment groups. Nevertheless, the histochemical investigation of calcium deposits revealed their widespread presence only in scaffolds seeded with differentiated cells (OM condition). 

Our analysis also highlighted a difference in distribution, dimension and intensity of calcium deposits under this condition. G-PEG-Dx1 better supported the FBS culture condition, while G-PEG-Dx2 fitted better with the hPL culture condition. G-PEG-Dx1 with FBS showed large calcium deposits in mashes with a medium intensity staining; interestingly, calcium deposits were detected only in scaffold mashes and not in extracellular matrix formed within the pores. On the contrary, G-PEG-Dx2 with hPL showed both large and very small calcium deposits with high intensity staining and widespread in all mashes. This new approach allowed us to correlate the chemical composition, pore dimensions and architecture of the scaffold with cell performance under specific stimuli (FBS and hPL). These data also suggest possible in vitro trophic effects of hMSCs [35,36] in inducing mineralization processes, that can be also influenced by the composition [37] or elasticity [38] of the scaffold. In addition, the scaffold porosity is fundamental to mimic the natural bone structure. Since human trabecular bone has a variable porosity that ranges between 40% and 95% of the bone volume [39,40], our scaffolds are adequate to reproduce the empty spaces of normal bone tissue, but G-PEG-Dx2 presents small pores that result in a wide surface area which is conducive to rapid calcium deposition. 

In the second step, we then investigated the composition of calcium deposits highlighted in optical microscopy. To this end, the mineralization process has been deeper examined both with SEM-EDX measuring the Ca P elements production and Raman micro-spectroscopy technique to monitor hydroxyapatite after 28 days of culture. 

EDX-SEM analysis highlights Ca P elements at the upper face of the sample as well as the penetration of the cellular culture inside the open porosity of the scaffold. These data were then confirmed with Raman spectroscopy. Few studies using Raman spectroscopy to study bone regeneration on bio-scaffolds are present in literature, even if this technique is the gold standard for a non-invasive and non-destructive analysis that provides real-time biological information about the sample [15].

Particularly, Chiang et al. [15] characterized a mineral marker during the osteoblast differentiation process by monitoring the Raman signal of hydroxyapatite. Recently, Du et al. [41] adopted Raman spectroscopy to explore the microstructure and chemical components of the new bone tissue in an animal model of bone defect. Moreover, Atachi et al. [42] evaluated the in vivo performance of a 3D scaffold specifically developed for bone tissue engineering also with the support of Raman spectroscopy. 

Here, we have confirmed that the 960 cm^−1^ can act as a main hydroxyapatite marker when monitoring osteoblasts during the differentiation process in accordance with previous study [15]. We also observed this signal in the center of the scaffold, where spherical particles can be observed. 

Matrix-assisted laser desorption/ionization mass spectrometry imaging (MALDI-MSI) was also employed to determine whether in situ lipidomics have the potential to provide further complementary, molecular evidence of osteogenic modulation in our 3D model. This represents a particularly stimulating aspect given that the technique facilitates in situ lipidomic analysis of individual cell-types within a complex biological background, such as regenerative scaffolds, and may highlight their implication in cell function and role [19,43]. Using this approach, distinct lipidomic signatures were generated from those hMSCs maintained in G-PEG-Dx1 and G-PEG-Dx2, the two scaffolds which presented the least and greatest differentiation in the previous analysis, respectively. In particular, *m*/*z* 885.59, related to PI (18:0/20:4), was found to be of higher abundance in G-PEG-Dx2 and is in accordance with the findings reported by Rocha et al., where this lipid species was associated with the latter stages of hBMSCs undergoing chondrogenesis and thus represents a promising indicator of osteogenic modulation. Moreover, the abundance of lipid species belonging to several classes, including glycerophosphoglycerols (PG), glycerophosphoinositols (PI) and glyceropyrophosphates (PPA), were found to be modulated as a result of the osteogenic differentiation process, supporting the growing body of evidence suggesting that phospholipid composition is altered during osteogenic differentiation [44,45]. Notwithstanding these promising preliminary findings, they should be investigated further in our planned future studies to confirm their relevance in the osteogenic process.

The results all together suggest that the mineralization process occurs not only in the surface but also in the middle part of the scaffold, highlighting an early mineralization process of the entire structure. 

## 5. Conclusions

In conclusion, our results demonstrated that both G-PEG-Dx1 and G-PEG-Dx2 synthetic hydrogel scaffolds are bio-active scaffolds able to support cell growth, osteo-differentiation and particularly mineralization, demonstrating their potential application for bone regeneration. We have shown that the morphological and physico-mechanical properties mediated in the same hydrogel composition by alterations in the crosslinking density (i.e., mechanical stiffness, architecture, pore sizes and hydrophilicity) must be considered in the context of the mineralization processes and osteogenic differentiation.

In particular, G-PEG-Dx2, combined with hPL and osteogenic factors, was more efficient than G-PEG-Dx1 in inducing BM-hMSCs osteogenic differentiation and promoting a faster mineralization process, suggesting a more favorable and active role for improving bone repair. Both Raman spectroscopy and MALDI-MSI provide solid objective evidence on the progress of differentiation, and our findings suggest that they may be able to provide additional molecular insights into the progress and nature of cell differentiation. In fact, the use of Raman spectroscopy enables a more specific signal to be obtained and is very important to our understanding and monitoring of osteoblast maturity during the differentiation process. Moreover, MALDI-MSI provides complementary molecular information and is capable of unravelling this lipidomic heterogeneity within the hMSCs seeded in the scaffold network, with lipid changes in these cells having already been shown to occur during cell differentiation and should be investigated to better understand the molecular nature of this process.

## Data Availability

Data sharing is not applicable to this article.

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
