# Peer review of "Mineralization of 3D Osteogenic Model Based on Gelatin-Dextran Hybrid Hydrogel Scaffold Bioengineered with Mesenchymal Stromal Cells: A Multiparametric Evaluation"

_materials, 2021, doi:10.3390/ma14143852_

Round 1
Reviewer 1 Report
In this manuscript the authors evaluate two hydrogel scaffolds based on gelatin and dextran: G-PEG-Dx1 and G-PEG-Dx2. These scaffolds have identical composition but different architecture, due to different degrees of crosslinking. Both hydrogels were seeded with bone marrow (BM) hMSCs and it was shown that while cell colonization is better in G-PEG-Dx1, G-PEG-Dx2 is more efficient in osteogenic differentiation when osteogenic factors are present.
The manuscript is well written and comprehensive, providing all necessary information to understand and repeat the experiment. I have one objection/suggestion on the study and particularly with the claim that MALDI-MSI is adequate in proving osteogenic differentiation. While the lipidomic analysis might be indicative of a phenotypic profile, it is by no means proof. The authors should explicitly state that this experiment does not constitute proof of osteogenic differentiation. Alternatively, they can include immunocytochemical stainings of cryosections of BM-hMSC laden scaffolds with osteogenic markers such as BMP-2, osteocalcin, and osteopontin.
Author Response
Thank you for your comments and advice. Following the kind suggestions of the Reviewer, the term “osteogenic differentiation” has been changed to “mineralization”. Please see page 1 lines 34, 40, 45; page 2 line 99, page 3 line 103; page 3 line 126, page 5 line 240, page 10 line 494, page 11 line 525.
Moreover, we have modified the statement in the introduction:” Other techniques typical of material science that was applied in literature for the study of mineralization and osteogenic differentiation of hMSCs is Raman spectroscopy (RS): this is a non-destructive technique, which measures the vibrational spectrum, allowing to discriminate between the chemical species. This means that RS allow to measure protein contribution of the cultured cells and the phosphate bands contribution to mineralization due to the presence of hydroxyapatite [14], with a spatial resolution of the order of a few microns and poor interference from water signal [15]. The microscopic investigation of non-flat surfaces by mean of optical microscopy of the Raman system suffers from low depth of focus typical of 50X objective, but is efficiently guided by the visualization of the scaffold surface and localization of mineral deposits obtained by SEM, and also from conventional staining method (Von Kossa).” Please see pages 2,3 lines 98-109.
Finally, we have also added the statement to the discussion on page 23, lines 713-715:” Notwithstanding these promising preliminary findings, they should be investigated further in our planned future studies to confirm their relevance in the osteogenic process”.
We hope that this satisfies the reviewer and underlines that this experiment alone does not constitute proof of osteogenic differentiation.

Reviewer 2 Report
Impressive, I read the paper several times, with no objections or recommendations. Congrats!
I recommend the publication in current form.
Author Response
Thank you very much for your appreciation to this work.
Reviewer 3 Report
This article shows potentially interesting data. However, some of the conclusions are not well supported by the data. The study is essentially based on the analysis of mineralization, but it is not clear from the presented data whether mineralization is really a cell-mediated process which depends on the differentiation status. Additional control experiments, which may have been performed but whose results are not shown, would strengthen the conclusions drawnw by the authors. Besides, the interpretation of the data obtained by MALDI-MSI analysis, which appears as an unconventional, potentially promising technique, are not validated by the appropriate control experiments.
Detailed comments:
- Cellular characterizations: The part of the manuscript dedicated to the cell studies is quite limited. It would have been very interesting to show the DAPI staining, which is obviously the only reliable technique, which in addition alows for quantification, for both gels. This would allow a comparison of the cell number/density and distribution in both gels. Moreover, this would be useful for the interpretation of the mineralization data. The comparison of cell distribution in different parts of the gel, in particular the central and the peripheral parts, could also be presented. This would also enable to correlate this distribution with the mineralization pattern.
- Analysis of mineralization. Does the presence of Von Kossa positive nodules correlate with the presence of cells? The authors mention that in growth medium there are few mineral deposits, but these data are not shown. However, I think this is an important information which should be shown in figure 4. More generally, I am not fully convinced that the mineralization is due to the presence of differentiated cells. This is very important to have strong evidences of this because in this study the osteoblastic differentiation phenotype of the MSC is entirely based on the presence of mineral. Therefore the data which can support this hypothesis must be provided (correlation between mineral nodules and cells, absence of nodules in cell-free gels, or lower number of nodules in growth medium....)
- Raman spectroscopy. This paragraph starts with the following sentence: "we studied the osteogenic differentiation of hMSCs by Raman spectroscopy". In fact this is the mineral crystals which are analyzed by this method, not the differentiated cells. Whether the minerals result from the differentiation of hBMSCs is not convincingly demonstrated in this study.
- MALDI-MSI analysis. This method is obviously powerful and can provide valuable information. However, as it is presented here, it only shows that there are different lipid profiles in the cells grown in the two gels. Whether this is related to differentiation remains speculative. I would have been very supportive to show the MALDI-MSI analysis of cells grown in GM (negative control providing the lipid profile of undifferentiated cells) , or of cells grown in OM in classical 2D method (positive control).
- The authors show differences between FBS and hPL in the mineralization pattern. However these differences are not the same in G-PEG-Dx1 and G-PEG-Dx2. What could account for this surprising discrepancy?
- 6. In the discussion about the MALDI-MSI data, what is the input of these results in the study? This analysis shows that a number of lipids are found at different levels in each of the cell-laden hydrogels. But these lipids are not those which are described in the previous studies as related to the osteoblast differentiation of MSCs. These findings, although potentially interesting, do not provide a significant support to the core of the manuscript. It would have been interesting for instance to analyze the lipidomics in a conventional culture under growth versus differentiation medium, to determine whether the lipidomics profile was similar to the one observed when MSCs are cultured in the hydrogels.
Author Response
Detailed comments:
- Cellular characterizations: The part of the manuscript dedicated to the cell studies is quite limited. It would have been very interesting to show the DAPI staining, which is obviously the only reliable technique, which in addition alows for quantification, for both gels. This would allow a comparison of the cell number/density and distribution in both gels. Moreover, this would be useful for the interpretation of the mineralization data. The comparison of cell distribution in different parts of the gel, in particular the central and the peripheral parts, could also be presented. This would also enable to correlate this distribution with the mineralization pattern.
We thank for the advice. As supplementary material we presented the cells distribution for each scaffolds and conditions. DAPI staining has to be considered as a qualitative analysis of the cells number in the scaffold. Both central and peripheral parts have been reported. Please see Supplementary Figure 1 page 25.
- Analysis of mineralization. Does the presence of Von Kossa positive nodules correlate with the presence of cells? The authors mention that in growth medium there are few mineral deposits, but these data are not shown. However, I think this is an important information which should be shown in figure 4. More generally, I am not fully convinced that the mineralization is due to the presence of differentiated cells. This is very important to have strong evidences of this because in this study the osteoblastic differentiation phenotype of the MSC is entirely based on the presence of mineral. Therefore the data which can support this hypothesis must be provided (correlation between mineral nodules and cells, absence of nodules in cell-free gels, or lower number of nodules in growth medium....).
We thank for the advice. As supplementary material we presented the figures of controls GM medium. Please see Supplementary Figure 2 page 25.
- Raman spectroscopy. This paragraph starts with the following sentence: "we studied the osteogenic differentiation of hMSCs by Raman spectroscopy". In fact this is the mineral crystals which are analyzed by this method, not the differentiated cells. Whether the minerals result from the differentiation of hBMSCs is not convincingly demonstrated in this study.
Our choice to refer to osteogenic differentiation (OD) derives from the observation that usually the mineralization is following osteogenic differentiation [Yorukoglu, A.C. et al. Stem Cells Int. 2017, https://doi.org/10.1155/2017/2374161]. We agree with the reviewer that Raman spectroscopy gives evidence of the presence of phosphate bands related to hydroxyapatite. Since control tests to confirm the OD are outside the scope of the present work, and will be performed in the future, we substituted “osteogenic differentiation” with “mineralization” to be consistent to what has been measured. Please see page 10 line 455.
We have also modified the statement: “We studied the mineralization of hMSCs by Raman spectroscopy, by following the Raman signal of hydroxyapatite in the polymeric scaffold.” Please see page 10 lines 493,494.
- MALDI-MSI analysis. This method is obviously powerful and can provide valuable information. However, as it is presented here, it only shows that there are different lipid profiles in the cells grown in the two gels. Whether this is related to differentiation remains speculative. I would have been very supportive to show the MALDI-MSI analysis of cells grown in GM (negative control providing the lipid profile of undifferentiated cells), or of cells grown in OM in classical 2D method (positive control).
The authors thank the reviewer for their comment regarding these additional experiments. While this may be an important aspect, the suggested work is outside the scope of this manuscript which was designed as a pilot study to highlight the feasibility of generating novel three-dimensional gelatin–dextran hybrid hydrogel scaffolds, that support mineralization, and characterizing their morphological and molecular characteristics.
In light of this comment, however, the authors have added a statement to the discussion on page 23, lines 713-715:” Notwithstanding these promising preliminary findings, they should be investigated further in our planned future studies to confirm their relevance in the osteogenic process” to underline the need for further, more detailed, lipidomics studies to confirm this relevance in the osteogenic differentiation process.
It is also important to note that, as referred to in the introduction, it can be somewhat challenging to characterize the lipidome in mesenchymal stem cells and the presented data, performed in novel regenerative scaffolds, may also be interesting from a technical standpoint, especially to those readers who may not be familiar with the technique.
- The authors show differences between FBS and hPL in the mineralization pattern. However these differences are not the same in G-PEG-Dx1 and G-PEG-Dx2. What could account for this surprising discrepancy?
We thank for your comments. There is a specific interaction between cells, the supplements added to their media and an individual scaffold. Particularly, hPL is known to have different behavior from FBS, thanks to the growth factor contained [Re, F et al., J. Tissue Eng. 2019, 10, 2041731419845852]. Consequently, hPL has shown to have different behavior to the mineralization pattern respect FBS as previous demonstrated [Re, F et al., J. Tissue Eng. 2019, 10, 2041731419845852; Doucet, C et al. J. Cell Physiol. 2005, 205, 228–236; Manferdini, C. et al. Biomaterials 2010, 31, 3986-3996.].
- In the discussion about the MALDI-MSI data, what is the input of these results in the study? This analysis shows that a number of lipids are found at different levels in each of the cell-laden hydrogels. But these lipids are not those which are described in the previous studies as related to the osteoblast differentiation of MSCs. These findings, although potentially interesting, do not provide a significant support to the core of the manuscript. It would have been interesting for instance to analyze the lipidomics in a conventional culture under growth versus differentiation medium, to determine whether the lipidomics profile was similar to the one observed when MSCs are cultured in the hydrogels.
The authors thank the reviewer for their comment. The inclusion of the MALDI-MSI data was to demonstrate the possibility of using this technique to detect different lipid compositions in the MSCs cultured in the different hydrogels, that may be associated with the degree of mineralization. The results presented here highlight this possibility, even in the somewhat challenging samples, and report lipid alterations that may be targeted and studied in greater detail in subsequent studies.
Please see modifications of the statements page 11, lines 522-528; page 23 lines 713-715; page 24 lines 734-736.
Interestingly, there is some overlap between our results and those observed by Rocha et al (https://doi.org/10.1002/pmic.201400260), who also used mass spectrometry imaging to characterize lipid markers of chondrogenic differentiation. The authors also note that it may not be surprising that, with different techniques, some diverse lipid alterations may be observed due to the inherent nature of each technique, which may favor the detection of certain classes of lipid molecules.

Reviewer 4 Report
The article entitled “Mineralization of 3D osteogenic model based on gelatin-dextran hybrid hydrogel scaffold bioengineered with mesenchymal stromal cells” represents a potential interesting study on the characterization of bioscaffolds in the field of tissue engineering.
However, the composition of the material is not novel, the authors have already published synthesis processes and characterizations of the bulk materials, and this current study does not offer novel casting or 3D printing procedures in order to mimic bone trabeculae or any other design, which instead would represent a great improvement in the area.
Therefore, I strongly suggest shifting the focus of the article unless the authors can produce data on 3Dprintability/casting techniques, characterization to mimic bone architecture or for personalized implant delivery for bone repair.
The hybrid hydrogel materials as well are not really suitable for bone implants unless mineralized matrix can accumulate and reach the mechanical properties of native bone, which is not observed in the current study. Nor the initial bulk stiffness nor the final one achieved after ECM deposition from MSC (mechanical data after osteogenesis not shown by the authors) is comparable to native bone, making these materials quite inadequate for bone repair purposes.
Another consideration regards the clinical translation: hybrid materials, despite showing greater improvement compared to the mono-composite counterpart in vitro, present challenges in terms of the clinical translation. The more additives are incorporated into materials and the more complicated synthesis are required to synthesis them and casting them, the more complicated regulatory constrains will delay or even kill the idea of using these materials into patient.
Nevertheless, I appreciate the effort of the authors to use different non-destructive characterization techniques (Raman, SEM-EDX, MALDI-MSI). I strongly recommend shifting the focus of this article on the workflow of analyses performed to evaluate the degree of osteogenesis, instead of the hybrid material itself. The authors should use well establish tissue engineered models of osteogenesis to correlate the extent of mineralization achieved using standard techniques (i.e. histology) with the non-destructive characterization analyses presented in the paper, and describe advantages and limitations of the characterizations used to evaluate the quality of a tissue engineered construct.
The need of using non-destructive and conservative analyses for 4D analysis of tissue engineering scaffolds is an emerging and crucial aspect in the field, as for instance well described in A.A. Appel, M.A. Anastasio, J.C. Larson, E.M. Brey, Imaging challenges in biomaterials and tissue engineering, Biomaterials 34(28) (2013) 6615-30
As already underlined in the Introduction section “A critical parameter to evaluate the tissue-engineered constructs is the identification of a quality control system to allow the qualitative and quantitative measurement of the mineralization area” this is the strength of this paper!
“One of them is Raman spectroscopy (RS), a technology able to perform real-time, nondestructive, and noninvasive measurements which provides vibrational information on the molecular structure of cells and their surroundings at micrometer resolution”
Additional comments:
The title is without doubt the part of a paper that is read the most, therefore I recommend changing the title according to the amended focus of the article, if the authors decide to resubmit the article following my suggestions.
Figure 1D: to standardize the type of mechanical analysis and make comparison with previous literature, the authors should show the calculated Compressive Young Modulus instead of the Elastic modulus [D. Loessner, C. Meinert, E. Kaemmerer, L.C. Martine, K. Yue, P.A. Levett, T.J. Klein, F.P.W. Melchels, A. Khademhosseini, D.W. Hutmacher, Functionalization, preparation and use of cell-laden gelatin methacryloyl-based hydrogels as modular tissue culture platforms, Nat. Protoc. 11 (2016) 727–746. http://dx.doi.org/10.1038/nprot.2016.037.]
Can the authors better define the equation used to calculate the elastic modulus and the precise range of the chosen slope?
Could the authors explain the profile of the curve: why there is a drop in compressive stress around 30% ?
It would be interesting to measure the mechanical properties after the osteogenesis, despite I’m not expecting any significant increase considering the rest of the results obtained.
Figure 2, 3, 4: the extend of mineralization is quite poor, minimal quantifications are shown, and clearly histological artefacts that derive from the harsh procedure itself are quite evident.
The cellular density used is quite low (106 cells/mL). I recommend repeating the experiments with high cell density at least 10x106 cell/mL, to provide cellular advantages in building mineralized matrix. In figure 3D is in fact quite evident that the cell density is really low.
“Von Kossa stain showed that the mineralization occurred only within scaffold meshes”. This is in line with other studies showing that cells entrapped in dense, non-degradable gels produce minimal extracellular matrix, which is confined to the space surrounding the cellular membrane, thus impairing a physiological development of new tissue. In degradable gels, on the other hand, the network density decreases with time while the mesh size increases, allowing for further matrix deposition and organization [Onofrillo et al, Biomaterials 2021 Jan;264:120383. doi: 10.1016/j.biomaterials.2020.120383.]
Therefore, this is further indication that the composition of the material used in the study is quite low permissive for mineralised matrix production and accumulation into the scaffolds.
Figure 6: What are we looking at ? what materials composition and conditions have been used for this analysis ? Which areas of the low magnification picture are reported in the lower panels? The SEM-EDX analysis is quite interesting, and it is my opinion that deserve more investigations and quantifications, despite the poor osteogenesis contribution of the model used could make the analysis more complicated.
General: I also recommend a grammatical and syntax revision of the English used through the whole manuscript. Several mistakes like the one at line 99 needs to be amended:
“In bones, however, hydroxyapatite crystals make it difficult [15] and this explains”…looks like part of the sentence is missing…
Also, I strongly recommend using the same style for the figures and graphs to make the work more professional.
Author Response
Comments and Suggestions for Authors
The article entitled “Mineralization of 3D osteogenic model based on gelatin-dextran hybrid hydrogel scaffold bioengineered with mesenchymal stromal cells” represents a potential interesting study on the characterization of bioscaffolds in the field of tissue engineering.
However, the composition of the material is not novel, the authors have already published synthesis processes and characterizations of the bulk materials, and this current study does not offer novel casting or 3D printing procedures in order to mimic bone trabeculae or any other design, which instead would represent a great improvement in the area.
We thank the reviewer for the comment and suggestions. We have previously shown that the presence of a polysaccharide such as chitosan in the composition of hybrid hydrogels positively affects the mineralization of mesenchymal cells. In this work we investigated novel composition containing dextran.
Furthermore, we have shown that the morphological and physico-mechanical properties mediated in the same hydrogel composition by alterations in the crosslinking density (i.e. mechanical stiffness, architecture, pores sizes and hydrophilicity) must be considered in the context of the mineralization processes and osteogenic differentiation.
In particular the lower water uptake, higher mechanical stiffness and lower pores sizes of G-PEG-Dx2, all indicating a higher cross-linking density than G-PEG-Dx1, resulted in an overall increase of Ca and P deposition and favored osteogenic differentiation. Please see the addition to the conclusions.
Therefore, I strongly suggest shifting the focus of the article unless the authors can produce data on 3Dprintability/casting techniques, characterization to mimic bone architecture or for personalized implant delivery for bone repair.
The hybrid hydrogel materials as well are not really suitable for bone implants unless mineralized matrix can accumulate and reach the mechanical properties of native bone, which is not observed in the current study. Nor the initial bulk stiffness nor the final one achieved after ECM deposition from MSC (mechanical data after osteogenesis not shown by the authors) is comparable to native bone, making these materials quite inadequate for bone repair purposes.
Another consideration regards the clinical translation: hybrid materials, despite showing greater improvement compared to the mono-composite counterpart in vitro, present challenges in terms of the clinical translation. The more additives are incorporated into materials and the more complicated synthesis are required to synthesis them and casting them, the more complicated regulatory constrains will delay or even kill the idea of using these materials into patient.
Nevertheless, I appreciate the effort of the authors to use different non-destructive characterization techniques (Raman, SEM-EDX, MALDI-MSI). I strongly recommend shifting the focus of this article on the workflow of analyses performed to evaluate the degree of osteogenesis, instead of the hybrid material itself.
We thank the reviewer for his/her comments. Hydrogel alone may be quite inadequate for general bone repair, but a hydroxyapatite-reinforced material is likely suitable for bone implants. Furthermore, the versatility of hybrid hydrogels can be useful in general for the realization of hybrid scaffolds in which the bioactive hydrogels are supported by stiffer synthetic biopolymers.
No doubt, selection of scaffolding materials is one of the most important considerations both in terms of physico-mechanical properties as well as to obtain FDA approval in the field of tissue engineering and regenerative medicine. Hybrid hydrogels are now being used for tissue regeneration to closely mimic ECM compositions and dynamics. Keep this in mind, in the present study, two natural polymers, namely, gelatin and dextran, and one synthetic polymer, PEG, have been used as scaffolding material. Importantly, gelatin, PEG and dextran have been already approved by the FDA (USA) as biocompatible materials in the various aspects of biomedical applications [see for example: Wang, Lei, et al. "Dual‐Functional Dextran‐PEG Hydrogel as an Antimicrobial Biomedical Material." Macromolecular bioscience 18.2 (2018): 1700325; and Mandal, Abhirup, et al. "Hydrogels in the clinic." Bioengineering & Translational Medicine 5.2 (2020): e10158].
The authors should use well establish tissue engineered models of osteogenesis to correlate the extent of mineralization achieved using standard techniques (i.e. histology) with the non-destructive characterization analyses presented in the paper, and describe advantages and limitations of the characterizations used to evaluate the quality of a tissue engineered construct.
The need of using non-destructive and conservative analyses for 4D analysis of tissue engineering scaffolds is an emerging and crucial aspect in the field, as for instance well described in A.A. Appel, M.A. Anastasio, J.C. Larson, E.M. Brey, Imaging challenges in biomaterials and tissue engineering, Biomaterials 34(28) (2013) 6615-30
As already underlined in the Introduction section “A critical parameter to evaluate the tissue-engineered constructs is the identification of a quality control system to allow the qualitative and quantitative measurement of the mineralization area” this is the strength of this paper!
“One of them is Raman spectroscopy (RS), a technology able to perform real-time, nondestructive, and noninvasive measurements which provides vibrational information on the molecular structure of cells and their surroundings at micrometer resolution”
We thank the reviewer for his/her detailed analysis of the scenario of tissue-engineered constructs, and the appreciation of our experimental approach to the use of non-destructive techniques. We take into account his suggestion of focusing more on the techniques used to investigate the degree of osteogenesis, by implementing a more detailed description of “material science” characterization techniques like (Raman, SEM-EDX, MALDI-MSI) that are coupled to other more conventional techniques in the bio-scaffold characterization in the Introduction section.
Please see the Introduction from line 99 to line 109:
“Other techniques typical of material science that was applied in literature for the study of mineralization and osteogenic differentiation of hMSCs is Raman spectroscopy (RS): this is a non-destructive technique, which measures the vibrational spectrum, allowing to discriminate between the chemical species. This means that RS allow to measure protein contribution of the cultured cells and the phosphate bands contribution to mineralization due to the presence of hydroxyapatite [14], with a spatial resolution of the order of a few microns and poor interference from water signal, [15]. The microscopic investigation of non-flat surfaces by mean of optical microscopy of the Raman system suffers from low depth of focus typical of 50X objective, but is efficiently guided by the visualization of the scaffold surface and localization of mineral deposits obtained by SEM, and also from conventional staining method (Von Kossa). “
Additional comments:
The title is without doubt the part of a paper that is read the most, therefore I recommend changing the title according to the amended focus of the article, if the authors decide to resubmit the article following my suggestions.
The title has been changed, as kindly suggested, in order to better clarify the aim of the work: “Mineralization of 3D osteogenic model based on gelatin-dextran hybrid hydrogel scaffold bioengineered with mesenchymal stromal cells: a multiparametric evaluation”.
Figure 1D: to standardize the type of mechanical analysis and make comparison with previous literature, the authors should show the calculated Compressive Young Modulus instead of the Elastic modulus [D. Loessner, C. Meinert, E. Kaemmerer, L.C. Martine, K. Yue, P.A. Levett, T.J. Klein, F.P.W. Melchels, A. Khademhosseini, D.W. Hutmacher, Functionalization, preparation and use of cell-laden gelatin methacryloyl-based hydrogels as modular tissue culture platforms, Nat. Protoc. 11 (2016) 727–746. http://dx.doi.org/10.1038/nprot.2016.037.]
Although the Elastic modulus corresponds to the Young Modulus of unidirectional tests such as the compression test, we have replaced Young Modulus instead of Elastic modulus.
Can the authors better define the equation used to calculate the elastic modulus and the precise range of the chosen slope?
The authors thank the reviewer for noting this. We have improved paragraph 2.3 with the following sentence: “Engineering stress (s) was calculated by dividing the recorded force by the initial cross-sectional area. Engineering strain (e) under compression was defined as the change in height relative to the original height of the freestanding specimen. The initial Young Modulus (stiffness) was calculated from the slope of the compressive stress–strain curves within the range of 5–10% strain. The compressive strength was defined as the stress at 50% strain.” Please see page 4 lines 188-192
Could the authors explain the profile of the curve: why there is a drop in compressive stress around 30% ?
The authors thank the reviewer for noting this. We have improved paragraph 3.1 with the following description: “The stress–strain compressive curves obtained for G-PEG-D1, and G-PEGD2 hydrogels clearly showed three distinct regions depending on the slope of the curve: i) linear elastic region due to the bending of pore walls and lamellae (0–20% strain), ii) collapsed plateau region with plain slope as a result of pore walls and lamellae buckling and yielding (20–40% strain), and iii) densification region with higher slope (strain-stiffening) due to the pore walls/lamellae crushing together (>40% strain).” Please see page 9 lines 405-411
It would be interesting to measure the mechanical properties after the osteogenesis, despite I’m not expecting any significant increase considering the rest of the results obtained.
The authors thank the reviewer for their comment regarding these additional experiments. While this may be an important aspect, the suggested work is outside the scope of this manuscript which was designed as a pilot study to highlight the feasibility of generating novel three-dimensional gelatin–dextran hybrid hydrogel scaffolds, that support mineralization, and characterizing their morphological and molecular characteristics. However, they should be investigated further in our planned future studies to confirm the mechanical properties after the osteogenesis.
Figure 2, 3, 4: the extend of mineralization is quite poor, minimal quantifications are shown, and clearly histological artefacts that derive from the harsh procedure itself are quite evident.
Thanks for this observation. We agree with the Reviewer but we have to consider that the mineralization generally occurred after 48 days, while our analysis have been performed after 28 days in order to evidence the early mineralization as in our previous work (Re, F et al., J. Tissue Eng. 2019, 10, 2041731419845852). Anyway, our 3D model should be investigated at different time points, e.g. 48 days, in our planned future studies to analyze the increase of mineralized areas.
The cellular density used is quite low (106 cells/mL). I recommend repeating the experiments with high cell density at least 10x106 cell/mL, to provide cellular advantages in building mineralized matrix. In figure 3D is in fact quite evident that the cell density is really low.
We thanks for the observation but unfortunately we can’t repeat these analysis for the time required to doing them. The seeding protocol has already been assessed (F.Re et al, 2019, Journal of Tissue Engineering). However, they should be investigated further in our planned future studies to provide cellular advantages in building mineralized matrix.
“Von Kossa stain showed that the mineralization occurred only within scaffold meshes”. This is in line with other studies showing that cells entrapped in dense, non-degradable gels produce minimal extracellular matrix, which is confined to the space surrounding the cellular membrane, thus impairing a physiological development of new tissue. In degradable gels, on the other hand, the network density decreases with time while the mesh size increases, allowing for further matrix deposition and organization [Onofrillo et al, Biomaterials 2021 Jan;264:120383. doi: 10.1016/j.biomaterials.2020.120383.] Therefore, this is further indication that the composition of the material used in the study is quite low permissive for mineralised matrix production and accumulation into the scaffolds.
We agree with the Reviewer but, as we previously described, we have to consider that the mineralization generally occurred after 48 days, while our analysis have been performed after 28 days in order to evidence the early mineralization.
Figure 6: What are we looking at ? what materials composition and conditions have been used for this analysis? Which areas of the low magnification picture are reported in the lower panels? The SEM-EDX analysis is quite interesting, and it is my opinion that deserve more investigations and quantifications, despite the poor osteogenesis contribution of the model used could make the analysis more complicated.
The authors thank the reviewer for his comment. Figure 6 shows the morphology of the polymeric scaffold and the evidence of the mineralization. The panoramic view (upper part of the figure) shows the top side of the cylindrical scaffold and the exposed longitudinal section. The Backscattered Electron imaging highlights a decoration of bright particles at the scaffold surface and in the open porosity of the scaffold as well. The view at high magnification (bottom-left of the picture) reveals that the bright particles are indeed aggregates of round particles which measure few microns in size and feature an elemental composition different than the surrounding polymeric scaffold / biological matrix. The EDX measured the elemental composition of these details, which were shown to contain primarily Ca and P, oxygen was also detected but the contribution of the underlying matrix could not be discriminated. The mapping of Ca and P dispersion in the sample is reported in the bottom part of the picture as blue/green colored images. The resulting maps are similar to the magnified SEM image and indicate that both Ca and P, i.e. the cations of hydroxyapatite, are characteristic elements of the bright particles, which can be considered as the result of the penetration of the cellular culture inside the scaffold and evidence of the mineralization process. Please see modified statement page 10 “SEM-EDX analysis”
General: I also recommend a grammatical and syntax revision of the English used through the whole manuscript. Several mistakes like the one at line 99 needs to be amended:
“In bones, however, hydroxyapatite crystals make it difficult [15] and this explains”…looks like part of the sentence is missing…
The authors thank the reviewer for noting this. While this was linked to the concept in the previous sentence, this has been modified to “In bones, however, hydroxyapatite crystals make it difficult to determine the distribution of biomolecules using MALDI-MSI [15] ….” For improved clarity.
Also, I strongly recommend using the same style for the figures and graphs to make the work more professional.
Thanks to the reviewers for nothing this. Figures 6 and 7 have been modified to uniform the style.

Round 2
Reviewer 1 Report
My comments have been addressed. The manuscript is ready to publish.
Reviewer 3 Report
The authors have answered all the queries raised in the previous review. The work is interesting and now deserves publication in "Materials", without further modifications.
Author Response
Thank you very much.
Reviewer 4 Report
I acknowledge the authors for responding to my comments, but I regret to inform them that I’m not satisfied with the answers provided. The major issue of this study is that the extend of mineralization is quite poor, minimal quantifications are shown, and clearly histological artefacts that derive from the harsh procedure itself are quite evident. Also, the new figure provided show no quantification and no mineralization (Supp Figure 2), the DAPI images (Supp Figure 1) do not provide evidence that the cells are still alive (what about metabolic or live and dead assays?), and I doubt they will make more mineralization deposition at day 48. The authors acknowledge “we have to consider that the mineralization generally occurred after 48 days, while our analysis have been performed after 28 days in order to evidence the early mineralization as in our previous work (Re, F et al., J. Tissue Eng. 2019, 10, 2041731419845852)” therefore unless they can provide the extended analysis, it is really difficult to convincingly demonstrate the hypothesis of their study.
As a consequence, their poor mineralization model impacts the judgement on the non destructive techniques used (Raman, SEM-EDX, MALDI-MSI) making quite hard to understand if they can depict differences in the degree of mineralization of the scaffolds, being that there is no mineralization happening.
The authors should use well establish tissue engineered models of osteogenesis to correlate the extent of mineralization achieved using standard techniques with the non-destructive characterization analyses presented in the paper, and describe advantages and limitations of the characterizations used to evaluate the quality of a tissue engineered construct.
Author Response
The mineralization has been quantified evaluating different parameters (such as percentage of positive area, intensity of signal and mean diameters of deposits). Moreover, it is difficult to understand how the Reviewer supposed histological artefacts. Considering that Supplemetary figure 2 clearly show no or very low calcium deposits using growth medium without osteogenic inducers. In fact, Supplementary figure 2 has been requested as negative control and represents the scaffold with cells cultured in medium without osteogenic inducers. On the contrary, when osteogenic inducers (OM medium) have been added, mineralization occurs. So, the results are in line with what was expected. In reply to Reviewer 3 we added DAPI images. No request about metabolic assay has been done. The techniques performed in these studies are able to detect different degrees of mineralization, even low expressed.
Anyway, we thank the Reviewer for the advice, which will be investigated further in our planned future studies.